# Eucalyptus-Mediated Synthesized Silver Nanoparticles-Coated Urinary Catheter Inhibits Microbial Migration and Biofilm Formation

**DOI:** 10.3390/nano12224059

**Published:** 2022-11-18

**Authors:** Sakkarin Lethongkam, Supakit Paosen, Siwaporn Bilhman, Krittima Dumjun, Suttiwan Wunnoo, Suntree Choojit, Ratchaneewan Siri, Chalongrat Daengngam, Supayang P. Voravuthikunchai, Tanan Bejrananda

**Affiliations:** 1Natural Product Research Center of Excellence, Faculty of Science, Prince of Songkla University, Hat Yai, Songkhla 90110, Thailand; 2Center of Antimicrobial Biomaterial Innovation-Southeast Asia, Prince of Songkla University, Hat Yai, Songkhla 90110, Thailand; 3Science for Industry Program, Faculty of Science, Prince of Songkla University, Hat Yai, Songkhla 90110, Thailand; 4Division of Biological Science, Faculty of Science, Prince of Songkla University, Hat Yai, Songkhla 90110, Thailand; 5Division of Physical Science, Faculty of Science, Prince of Songkla University, Hat Yai, Songkhla 90110, Thailand; 6Department of Surgery, Division of Urology, Faculty of Medicine, Prince of Songkla University, Songkhla 90110, Thailand

**Keywords:** urinary tract infection, silver nanoparticles, foley catheter, biofilm, antimicrobial coating, *Eucalyptus camaldulensis*

## Abstract

Catheter-associated urinary tract infections (CAUTIs) are significant complications among catheterized patients, resulting in increased morbidity, mortality rates, and healthcare costs. Foley urinary catheters coated with synthesized silver nanoparticles (AgNPs) using *Eucalyptus camaldulensis* leaf extract were developed using a green chemistry principle. In situ-deposited AgNPs with particle size ranging between 20 and 120 nm on the catheter surface were illustrated by scanning electron microscopy. Atomic force microscopy revealed the changes in surface roughness after coating with nanoparticles. The coated catheter could significantly inhibit microbial adhesion and biofilm formation performed in pooled human urine-supplemented media to mimic a microenvironment during infections (*p* 0.05). AgNPs-coated catheter exhibited broad-spectrum antimicrobial activity against important pathogens, causing CAUTIs with no cytotoxic effects on HeLa cells. A reduction in microbial viability in biofilms was observed under confocal laser scanning microscopy. A catheter bridge model demonstrated complete prevention of *Proteus mirabilis* migration by the coated catheter. Significant inhibition of ascending motility of *Escherichia coli* and *P. mirabilis* along the AgNPs-coated catheter was demonstrated in an in vitro bladder model (*p* 0.05). The results suggested that the AgNPs-coated urinary catheter could be applied as an alternative strategy to minimize the risk of CAUTIs by preventing bacterial colonization and biofilm formation.

## 1. Introduction

Catheter-associated urinary tract infections (CAUTIs) are among the most common bacterial infections acquired both in hospitals and nursing homes. Patients with this infection are likely to have a high risk for bacteremia, a serious complication leading to increased morbidity and mortality [1]. The infections can occur within a few days of catheterization, as the urinary catheter facilitates the migration of microorganisms from the environment into the urinary bladder whilst impairing the natural antimicrobial mechanisms of the urinary tract. Moreover, urinary catheters also allow microbial adhesion and biofilm formation, resulting in the occurrence of drug-resistant microorganisms and difficult-to-treat infections. The most common causative pathogens causing CAUTIs are Gram-negative bacteria including *Escherichia* spp. and *Enterobacter* spp. More than 70% of these microorganisms are multidrug-resistant isolates [2].

Among antimicrobial agents, silver nanoparticles (AgNPs) are widely reported as an effective antimicrobial compound against various species of bacteria and fungi including multidrug-resistant microorganisms. Several approaches for the synthesis of AgNPs have been reported. A synthesis method using plant extracts as reducing and stabilizing agents is increasing in interest as it is simple, environmentally friendly, and inexpensive. Our previous studies demonstrated that the green synthesis of AgNPs using leaf extracts from plants in the Myrtaceae family, such as *Eucalyptus* sp., present strong antimicrobial activity [3,4]. *Eucalyptus camaldulensis* is one of the most common species, because it is widely cultivated in many countries, including Thailand, for paper industries. In addition, *E. camaldulensis* leaf extract is composed of several phytochemicals, such as polyphenols, carboxylic acids, and proteins, that may help in reducing Ag^+^ to Ag^0^ [4]. Green synthesized AgNPs have been used as an antimicrobial coating compound for many medical devices including endotracheal tubes, titanium implants, catheters, surgical sutures, and textiles [5,6,7,8,9]. Material surfaces have been modified by fabricating with AgNPs using different approaches including chemical, physical, and sputtering deposition methods. However, an in situ method for depositing AgNPs on material surfaces is a facile procedure that has been wildly reported to overcome the complicated multistep techniques.

The current study proposes a dual-functional antimicrobial migration and antibiofilm coating for urinary catheters. The method was developed using the principles of green chemistry. *E. camaldulensis*-mediated synthesized silver nanoparticles were allowed to form inside the substrate and on the catheter surfaces. The AgNPs-coated urinary catheter could inhibit important microorganisms causing CAUTIs. Biofilm formation was substantially reduced on the coated surface. An in vitro model simulating urinary tract infections caused by catheterization showed that AgNPs-coated catheters could prevent the migration of the tested pathogens. No cytotoxic effects on human cell lines were observed. The results suggested that green-synthesized AgNPs-coated urinary catheters may reduce the chance of getting CAUTIs.

## 2. Materials and Methods

### 2.1. Materials

Silver nitrate (99.9999%) was purchased from Sigma-Aldrich (St. Louis, MO, USA). Microbial reference strains were obtained from the American Type Culture Collection (Manassas, VA, USA). Clinical isolates were collected from Songklanagarind hospital (Ethical Approval No. REC 63-073-10-1). Culture media for microorganisms was obtained from Becton, Dickinson and Company (Sparks, MD, USA). Dulbecco’s modified Eagle’s medium (DMEM), fetal bovine serum, and trypan blue stain were purchased from Gibco Laboratories (Grand Island, NY, USA). Twenty-four well and 96-well cell culture plates were obtained from SPL Life Science (Gyeonggi-do, Republic of Korea). All other chemicals were purchased from Merck KGaA (Darmstadt, Germany) unless otherwise specified. Commercial silicone-coated latex Foley catheters were used in this study. *Eucalyptus camaldulensis* leaves were collected from Kaeng Khoi, Saraburi, Thailand, in April 2017. The leaves were authenticated by Dr. Phattaravee Prommanut. A voucher specimen (BK No. 070170) was deposited in the Bangkok Herbarium, Bangkok, Thailand [10].

### 2.2. Coating Process

Foley urinary catheters were cut into 0.5 cm segments. The segments were rinsed with deionized water and then immersed in silver salt solution (0.1 g/mL). Following incubation for 5 days under a dark condition, the samples were washed twice with deionized water, and *E. camaldulensis* extract was added, left at room temperature for a further 24 h. To obtain the AgNPs-coated urinary catheters, the samples were removed from the extract, washed with deionized water, and air-dried.

### 2.3. Characterization of AgNPs-Coated Urinary Catheter

Cross-sections of the coated urinary catheter were subjected to line scans using energy-dispersive X-ray spectroscopy (EDX) (Quantax 70, Hitachi, Japan) to determine the presence of silver elements on the top surface and inside the tube materials. Size distribution of AgNPs on the coated surface was determined based on a scanning electron microscope micrograph taken at magnification values of 150,000× by field emission-scanning electron microscope (FE-SEM) (Quanta 400, FEI, Eindhoven, The Netherlands), and the size distribution of the particles was determined by ImageJ. Surface roughness and topology were examined at the representative areas of the coated surface (5 × 5 mm) by using a noncontact mode of atomic force microscopy (AFM) (easyScan 2, Nanosurf, Switzerland).

### 2.4. Preparation of Pooled Human Urine

To simulate biological conditions, antimicrobial activities of the coated catheter were tested in a culture medium supplemented with sterile pooled human urine. Human urine was collected from three to ten healthy men and women volunteers who had no history of urinary tract infections or antibiotic use in the previous 2 months. The urine was pooled, filter-sterilized, stored at 4 °C, and used within the following 2–3 days. Informed consent was obtained from all volunteers. The protocol was approved by the Human Research Ethics Committee, Faculty of Medicine, Prince of Songkla University (Protocol No. 63-073-10-1).

### 2.5. Agar Inhibition Assay

A modified Kirby–Bauer method was performed in order to measure the inhibition zones of the AgNPs-coated urinary catheters against pathogenic microorganisms. Mueller Hinton agar (MHA) was inoculated with the test organisms mentioned above. The coated and uncoated urinary catheters were cut into 0.5 cm-long segments and embedded in the inoculated MHA. The inhibition zones were measured after incubating the MHA plates at 37 °C for 24 h. The diameter of the clear zone was measured using a vernier caliper, which is precise up to two decimal places. The results were mean ± standard deviation (SD) from three independent experiments performed in triplicate.

### 2.6. Antimicrobial Adhesion

Anti-planktonic growth and antimicrobial adhesion of the AgNPs-coated urinary catheters were tested against important pathogens causing CAUTIs including *Enterococcus faecalis* ATCC 29212, *Staphylococcus aureus* ATCC 25923, *Staphylococcus epidermidis* ATCC 35984, *Staphylococcus saprophyticus* NPRCoE 192201, *Escherichia coli* ATCC 25922, *Klebsiella pneumoniae* ATCC 700603, *Proteus mirabilis* NPRCoE 192201, *Pseudomonas aeruginosa* ATCC 27853, and *Candida albicans* ATCC 90028. The catheter segments were immersed in microbial suspension and incubated at 37 °C for 24 h. After that, sample segments were removed from the cultures and washed twice with phosphate-buffered saline (PBS) to remove nonadherent cells. To determine the anti-adhesion effects, the attached microorganisms on urinary catheter surfaces were extracted by placing the samples in a 0.9% saline solution, followed by sonication for 15 min, and vortex for 1 min. The microbial cells in the solution were 10-fold serially diluted and plated on agar. Colonies were counted after incubation at 37 °C for 24 h.

### 2.7. Antibiofilm Formation

The effects of the coated urinary catheter on biofilm formation of *S. aureus* ATCC 25923 and *E. coli* ATCC 25922, a representative strain of Gram-positive and Gram-negative bacteria, respectively, were determined using a 3-(4,5-dimethylthiazol-2-yl)-2,5-diphenyltetrazolium bromide (MTT) reduction assay and confocal laser scanning microscope (CLSM). AgNPs-coated catheters were challenged with 10^6^ CFU/mL of *S. aureus* ATCC 25923 or *E. coli* ATCC 25922 in culture-media-supplemented pooled human urine. After incubation for 24 h, the coated samples were washed twice with PBS to remove the unattached cells. Microbial cell viability in biofilms was observed using MTT assay. The presence of microorganisms on material surfaces was observed under CLSM after staining the materials with a LIVE/DEAD^®^ viability kit.

### 2.8. A Catheter Bridge Model

The model was constructed on tryptone soya agar with two 0.85 cm-wide perpendicular channels. Ten microliters of the bacterial cultures (10^8^ CFU/mL) were inoculated at the edge of the central channel of each plate and dried before the 1.0 cm-long catheter segments were placed across the channels. After 24, 48, and 72 h of incubation at 37 °C, the growth of test strains on the uninoculated agar was examined.

### 2.9. A Modified in Vitro Bladder Model

A simple in vitro bladder model was modified according to a previously report [11]. The catheter segments (3.5 cm long) were perpendicularly immersed in the bacterial suspension (10^8^ CFU/mL, 5% dextrose solution). The end of the catheter (0.5 cm long) immersed in the bacterial suspension was regarded as the site of infection. After 7 days’ coincubation, the catheters were removed from the suspension; microbial cell migration to the catheter surface was determined using the method as described in antimicrobial adhesion.

### 2.10. Cytotoxicity Test

Cytotoxicity of the AgNPs-coated urinary catheter was carried out according to ISO 10993-5. Human cervical cancer (HeLa) cells were incubated with extracts solution from the AgNPs-coated urinary catheter. The extracts were obtained by immersing the coated catheter in a DMEM medium supplemented with pooled human urine and incubated at 37 °C for 24 h. Cell viability after the treatment was determined by a cell metabolic activity assay by measuring optical density at 570 nm.

### 2.11. Ethical Approval

Our experiments on humans and/or the use of human tissue samples were performed in accordance with relevant guidelines and regulations and we confirmed that informed consent was obtained from all subjects and/or their legal guardians. The Songklanagarind hospital Ethics Committee granted approval for this study in accordance with the Declaration of Helsinki’s guiding principles (REC- 63-073-10-1).

## 3. Results and Discussion

### 3.1. Surface Characterization

The present development demonstrated a nano-modified surface of a Foley catheter by fabricating the surface with eucalyptus-mediated synthesized AgNPs using an in situ approach. The coating method allowed the deposition and formation of AgNPs both inside and on the surface of materials. An increase in surface roughness was observed under AFM, as demonstrated in Figure 1.

An uncoated catheter showed a smooth surface with root mean square of 78 nm, while an AgNPs-coated Foley catheter demonstrated a surface roughness of approximately 199 nm. The presence of AgNPs on the material surface and inside the catheter was confirmed using FESEM and EDX (Figure 2). EDX elemental color mapping images of the coated catheter was demonstrated in Appendix A. AgNPs were found both on the surfaces and inside the materials. FESEM illustrated small-size particles deposited on the coated catheter surface with particle size distribution ranging between 20 and 120 nm (Figure 3). The highest Ag peak was observed on the surface of the coated catheter. The synergy of both impregnated AgNPs inside catheters and the presence of nano-roughness on the surface strengthens the antimicrobial activities of the coated urinary catheter, making it more difficult for microorganisms to attach and colonize. A surface topology represents one of the most important factors for bacterial colonization. Surfaces with nano-scale roughness can prevent microbial adhesion. Small pits and canyons of small size could restrict flagellar rotation and limit bacterial dispersion, resulting in the reduction of microbial colonization and biofilm formation [12].

### 3.2. Antibacterial Activity of the AgNPs-Coated Urinary Catheter

Inhibition zones were determined by the measurement of the inhibition zone diameter. The AgNPs-coated urinary catheter demonstrated inhibition zones ranging between 8.98 and 10.58, 7.47 and 13.28, and 8.82 mm for Gram-negative, Gram-positive bacteria, and fungi, respectively (Table 1). A modified Kirby–Bauer method has been extensively used to screen the antibacterial activity of several antimicrobial compounds and antimicrobial-coated medical devices. A silicon catheter coated with *Pistacia lentiscus*-mediated synthesized AgNPs could inhibit both Gram-positive and Gram-negative bacteria [13]. Similarly, others have demonstrated a bio-inspired antimicrobial coating against *S. aureus* and *E. coli* [14].

### 3.3. Antimicrobial Adhesion

To simulate the biological conditions, the antimicrobial adhesion of AgNPs-coated urinary catheter was performed in culture media supplemented with pooled human urine against important pathogens causing CAUTIs. Following 24 h of incubation, the coated catheters significantly reduced the adhesion of all the tested microorganisms by approximately 3 log CFU/mL, compared with the uncoated urinary catheters (*p* 0.05) (Figure 4). The emergence and rapid spread of multidrug-resistant microorganisms have become a worldwide public health problem, particularly for medical device-associated infections, as the materials used to support patient life are also promoting microbial adhesion and biofilm formation. Pathogens within biofilms are troublesome to treat because the biofilm matrix can protect microorganisms from antimicrobials and host defense mechanisms. Indwelling medical devices with antimicrobial adhesion have been presented as an alternative preventive strategy to reduce the risk of infections. Silver nanoparticles have emerged as a suitable antimicrobial compound since they are able to overcome multidrug-resistant organisms. The nanoparticles provide potential advantages over conventional antibiotics for the treatment of bacterial infections by demonstrating a multi-target mode of action against bacterial cells causing difficulty in the adaptation of bacteria to develop resistance to AgNPs [15].

The AgNPs-coated Foley catheter demonstrated pronounced antimicrobial activity by preventing bacterial and fungi attachment to the surface. The results were achieved by the action of both AgNPs and Ag^+^. Silver nanoparticles could release Ag^+^ to bind with the thiol group at an active site of some proteins, resulting in inhibition of enzyme activity [16]. Silver ions inhibit activities of several enzymes involved in various important metabolic pathways of bacteria including the glycolysis pathway, oxidative stress homeostasis, and the pentose phosphate pathway, resulting in cell death [15]. In addition to the killing activity of Ag^+^ releasing from AgNPs, the coated catheter could prevent bacterial adhesion by repulsive forces of the AgNPs. The eucalyptus-mediated synthesized AgNPs used in this study presented an overall negative charge on their surfaces [4]. Meanwhile, most microbial cells possess a negative charge on their cell wall. The similarity of the presence of a net negative electrostatic charge of both the coated catheter and microorganisms resulted in antibacterial adhesion of the AgNPs-coated urinary catheter.

### 3.4. Antibiofilm Formation

The urine of healthy individuals is normally less favorable for bacterial growth because of a limitation of nutrients, low pH, high concentration of urea, and the presence of some proteins that act as antimicrobials such as Tamm–Horsfall glycoprotein [17] and siderocalin [18]. Changes in the urinary composition from various conditions including diabetes [19], diet [20], and inflammation [21] can facilitate bacterial growth and may affect the susceptibility of patients to infections. In addition, some pathogens are able to adapt themselves to survive in the hostile condition of urine. *E. coli* produces enterobactin and aerobactin for iron utilization in order to promote growth in iron-limited human urine [22]. A nickel transporter (Nik) of *S. aureus* is necessary for nickel uptake and nickel-dependent urease. The Nik component has been proposed as an important virulence factor for *S. aureus* during urinary tract infections [23]. A morphological change of *C. albicans* from yeast to hyphae was observed during growth in a catheterized environment [24].

Therefore, the antibiofilm activity of AgNPs-coated Foley catheters was investigated in media supplemented with pooled human urine against representative microorganisms. After co-incubation for 24 h, the cell viability in biofilms of the tested microorganisms on the coated catheters was reduced by up to 85%, compared with the uncoated urinary catheters (Figure 5). Visualization of biofilms after co-incubation was observed under CLSM, as shown in Figure 5B. The photographs illustrate the reduction in biofilms on AgNPs-coated urinary catheters of all the tested organisms including *S. aureus*, *E. coli*, *P. aeruginosa*, and *C. albicans*. Dense viable cells in biofilm within the biofilm thickness were clearly observed on the uncoated catheters, while less green fluorescence was apparent on the coated catheters with regard to both microorganisms. The results confirmed that AgNPs-coated catheters could potentially prevent the formation of biofilm with regard to the tested pathogens.

### 3.5. Antimicrobial Migration

Catheter-associated urinary tract infections are usually identified as polymicrobial infections because of the presence of biofilms on material surfaces. *P. mirabilis* was chosen for this investigation as they are the most common bacteria isolated from polymicrobial CAUTIs and mostly found in long-term catheterized patients [25]. The ability of the swarming motility of this pathogen has been noted as a unique and important virulence factor that can facilitate the migration across, and the colonization on Foley catheters of other nonmotile microorganisms. [26].

The catheter bridge model was constructed with regard to investigating the antibacterial migration of AgNPs-coated urinary catheters. The results are demonstrated in Figure 6. After incubation for 24 h, *P. mirabilis* could migrate through the uncoated catheters. The distance area and colony size were gradually increased throughout the test period. In contrast, the organisms could not migrate through the coated catheters. This could be explained by the presence of AgNPs on the material surface, which could block microorganisms from traveling from one side to another. However, the model might not exactly mimic real conditions during infections since the catheters were attached between solid agar without a flow of fluid over the material surfaces. To mimic the environment of CAUTIs, the antimicrobial migration of the coated catheters was further investigated in a modified in vitro bladder model [11]. After incubation for 7 days, a significant difference in microbial cells between AgNPs-coated and uncoated urinary catheters was observed (*p* 0.05). The highest numbers of the tested microorganisms at the infection site were observed on both the coated and uncoated urinary catheter surfaces. However, a significant reduction in the *E. coli* and *P. mirabilis* migration on the coated catheters was determined at 2 and 3 cm from the infection site, respectively, compared with the uncoated catheters (Figure 7).

Catheter-associated urinary tract infections can be caused by intraluminal or extraluminal infections. The catheter bridge model was constructed to mimic intraluminal infection, while the in vitro bladder model could represent both intraluminal and extraluminal infections. The occurrence of an intraluminal infection may be due to the contamination of the urine collection bag and subsequent ascension to the urinary bladder. Microorganisms in the environment and the normal flora of patients can colonize and form a biofilm on the surface of the Foley catheter and then move along the catheter, entering the bladder and causing an extraluminal infection [27]. The AgNPs-coated urinary catheters demonstrated the potential effects of inhibiting the bacterial migration of both intraluminal and extraluminal models of infection.

### 3.6. Cytotoxicity

Cytotoxic effects of antimicrobial agents released from anti-infective medical devices possess a major concern for novel biomaterials. The biocompatibility of AgNPs-coated catheters was investigated against a mammalian cell line using a metabolic activity assay. HeLa cells were incubated with the extraction medium obtained from the coated catheter after 24, 48, and 72 h of incubation. The AgNPs-coated catheter was highly biocompatible on HeLa cells, but the percentage of cell viability remained at 100%, compared with the untreated control (Figure 8). The results indicated that the silver ions released from the AgNPs-coated urinary catheter were not toxic for the mammalian cells.

## 4. Conclusions

Eucalyptus-mediated synthesized AgNPs-coated Foley urinary catheters provided nano-rough surfaces. The biosynthesized AgNPs were found both on the catheter surfaces and inside the material substrate. AgNPs-coated Foley catheters exhibited broad-spectrum antimicrobial activity against important pathogens causing CAUTIs. The coated catheter demonstrated as being effective in the culture medium supplemented with human urine. There were no cytotoxic effects of the coated catheters on HeLa cells. In vitro models mimicking the pathogenesis of CAUTIs revealed that the coated urinary catheters were able to inhibit bacterial migration from the contaminated sites. The promising results demonstrated that AgNPs-coated urinary catheters could be applied to catheterized patients to prevent microbial adhesion, a vital step for developing infections.

## Figures and Tables

**Figure 1 nanomaterials-12-04059-f001:**
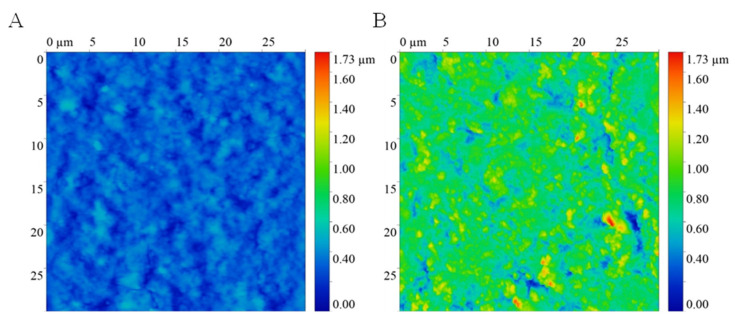
Atomic force microscopy micrographs of (**A**) uncoated and (**B**) AgNPs-coated urinary catheters.

**Figure 2 nanomaterials-12-04059-f002:**
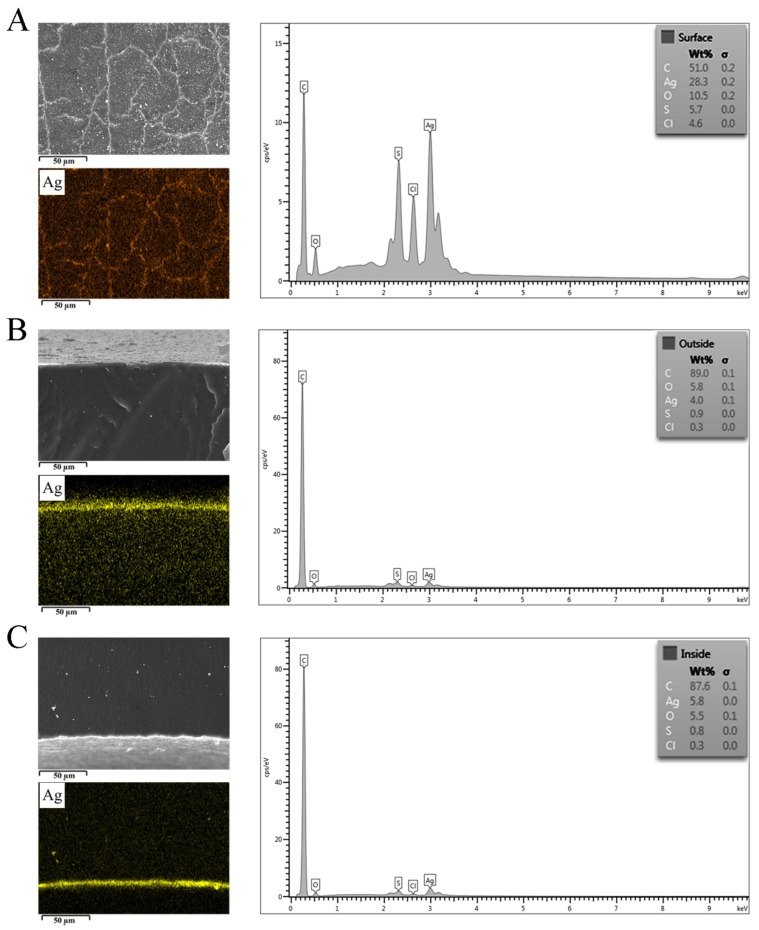
Field emission scanning electron microscope micrographs and energy-dispersive X-ray spectroscopy elemental mapping of (**A**) surface, (**B**) cross-sectional outer and (**C**) inner surfaces of AgNPs-coated urinary catheters.

**Figure 3 nanomaterials-12-04059-f003:**
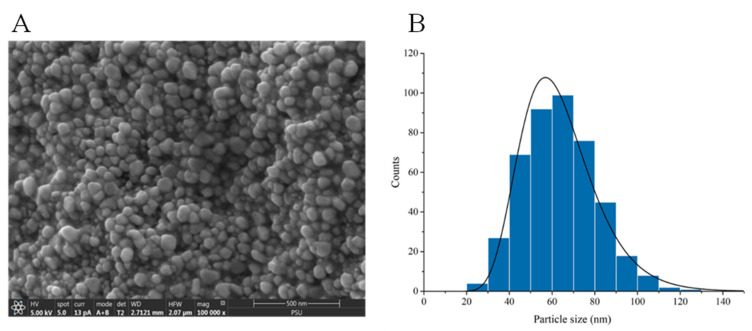
(**A**) Representative scanning electron microscopy image of AgNPs-coated urinary catheters and (**B**) particle size distribution of AgNPs on the coated surface.

**Figure 4 nanomaterials-12-04059-f004:**
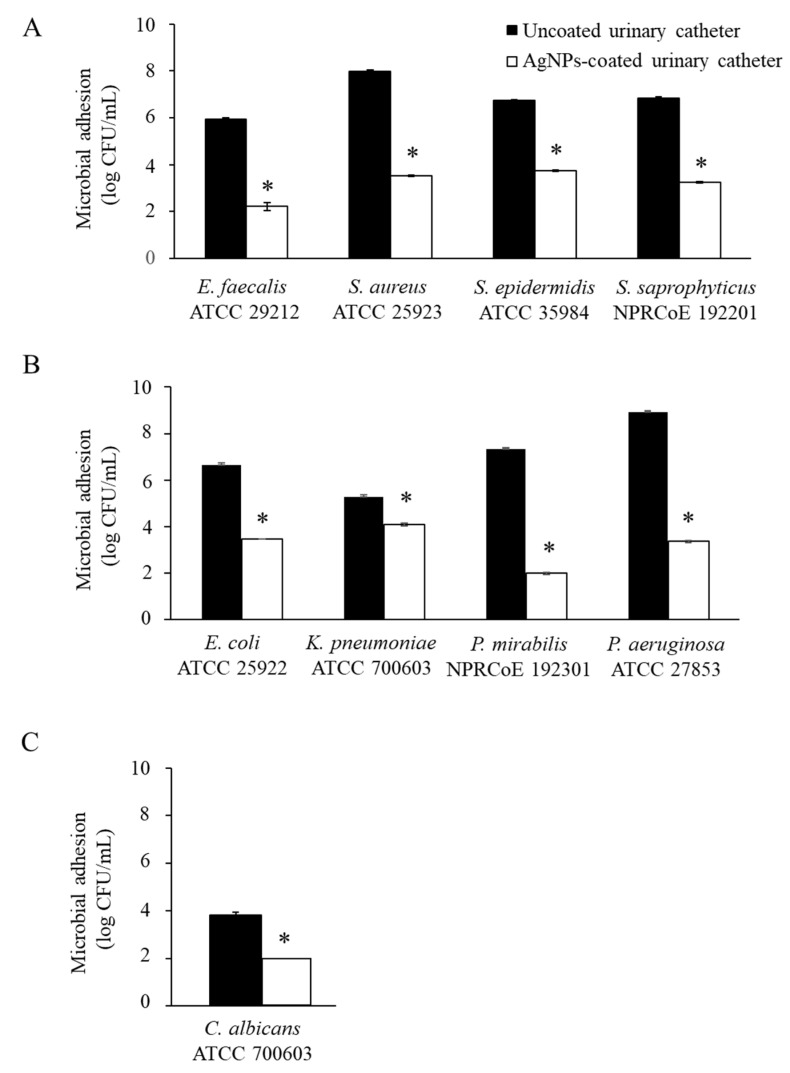
Antimicrobial adhesion following 24 h incubation of uncoated and AgNPs-coated urinary catheters against (**A**) Gram-positive bacteria, (**B**) Gram-negative bacteria, and (**C**) fungi. The values indicate mean ± SD from three independent experiments performed in triplicate, * *p* 0.05.

**Figure 5 nanomaterials-12-04059-f005:**
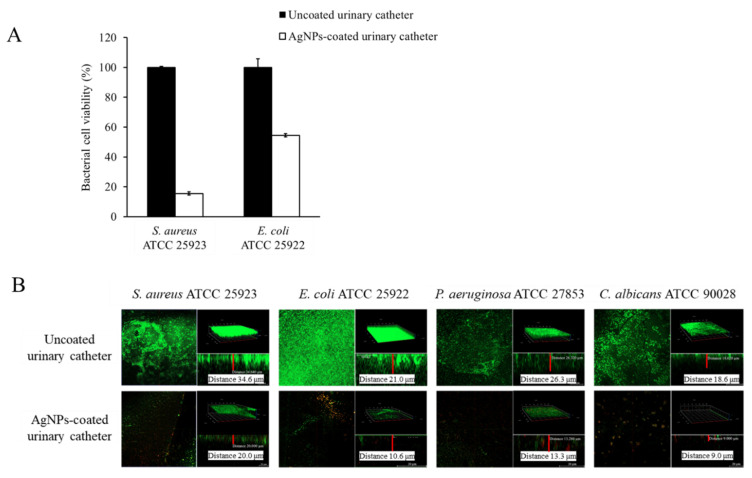
Effects of AgNPs-coated urinary catheter on biofilm formation of *Staphylococcus aureus* ATCC 25923 and *Escherichia coli* ATCC 25922 following incubation for 24 h. (**A**) Cell viability in biofilms was determined using MTT assay. The values indicate mean ± SD from three independent experiments performed in triplicate, *p* 0.05. (**B**) Biofilms were observed under confocal laser scanning microscopy after staining with LIVE/DEAD^®^ viability fluorescent dye. A representative photograph was from one of the three independent examinations.

**Figure 6 nanomaterials-12-04059-f006:**
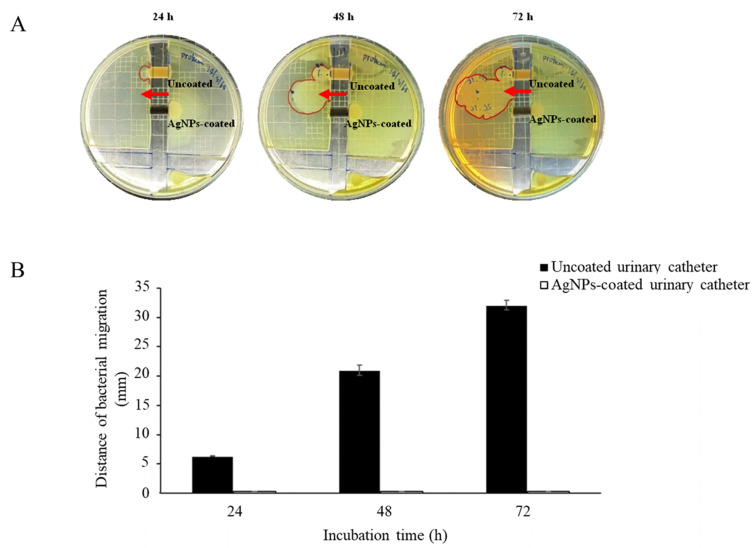
(**A**) Catheter bridge model of *Proteus mirabilis* migrating over 1 cm sections of uncoated and AgNPs-coated urinary catheter after 24, 48, and 72 h. The arrow demonstrates bacterial migration from the right to left side. (**B**) The distance of the bacterial migration values. The values indicate mean ± SD from three independent experiments performed in triplicate, *p* 0.05. A representative photograph was from one of the three independent examinations.

**Figure 7 nanomaterials-12-04059-f007:**
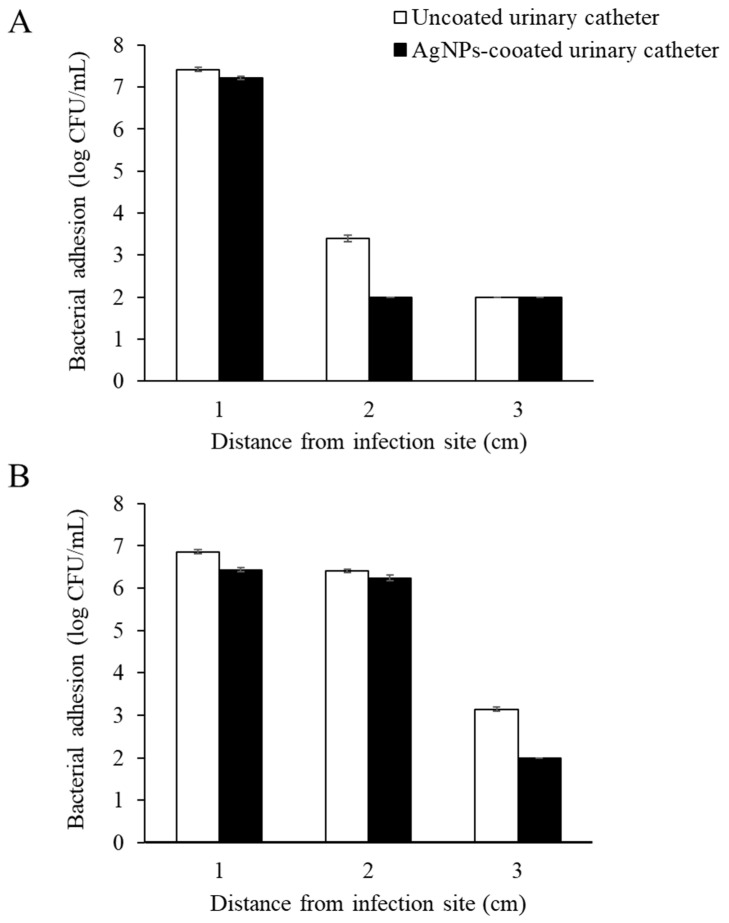
Numbers of microbial cell adhesion to different sections of uncoated and AgNPs-coated urinary catheters following 7 days of co-incubation with (**A**) *Escherichia coli* and (**B**) *Proteus mirabilis*. The values indicate mean ± SD from three independent experiments performed in triplicate, *p* 0.05.

**Figure 8 nanomaterials-12-04059-f008:**
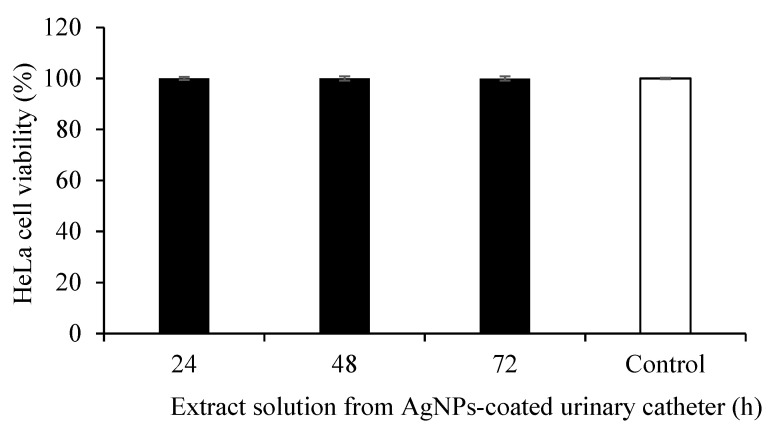
Cytotoxicity of AgNPs-coated urinary catheter on human cervical cancer (HeLa) cells. Cells were treated with extracted medium from AgNPs-coated urinary catheter after 24, 48, and 72 h. The values indicate mean ± SD from three independent experiments performed in triplicate, *p* 0.05.

**Table 1 nanomaterials-12-04059-t001:** Inhibition zone of AgNPs-coated urinary catheter against pathogenic microorganisms.

Microorganisms	Inhibition Zone (mm)	
	Uncoated Urinary Catheters	AgNPs-Coated Urinary Catheters
**Gram-positive bacteria**		
*Enterococcus faecalis* ATCC 29212	- ^a^	10.7 ± 0.3 ^b^
*Staphylococcus aureus* ATCC 25923	-	10.6 ± 0.1
*Staphylococcus epidermidis* ATCC 35984	-	9.0 ± 0.4
*Staphylococcus saprophyticus* NPRCoE 192201	-	9.7 ± 0.7
**Gram-negative bacteria**		
*Escherichia coli* ATCC 25922	-	8.0 ± 0.6
*Klebsiella pneumoniae* ATCC 700603	-	12.3 ± 0.9
*Proteus mirabilis* NPRCoE 192201	-	7.5 ± 0.4
*Pseudomonas aeruginosa* ATCC 27853	-	13.3 ± 0.2
**Fungi**		
*Candida albicans* ATCC 90028	-	8.8 ± 0.1

^a^ no zone; ^b^ the results were means ± SD from three independent experiments performed in triplicate.

## Data Availability

The data that support the findings of this study are available from the corresponding author upon reasonable request.

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
