# Peer review of "Eucalyptus-Mediated Synthesized Silver Nanoparticles-Coated Urinary Catheter Inhibits Microbial Migration and Biofilm Formation"

_nanomaterials, 2022, doi:10.3390/nano12224059_

Round 1

Reviewer 1 Report

In the submitted manuscript, the authors have proposed a green synthesis method of Ag NPs on urinary catheters and have characterized them thoroughly, with special focus on the antimicrobial properties. The work is well suited for the journal and interesting enough for its publication but there are several issues to be addressed before I recommend its publication. In particular, my commentaries are the following ones:

a) There are several grammar errors and/or phrases which meaning is not totally clear. Please, check the grammar carefully. I put here several examples.

i)                   Line 55-56: “…have been modified by fabricating with AgNPs using different approaches…” I think that the “with” should be omitted.

ii)                 Line 81: I am not sure of the meaning of “…balloon catheter was used silicone-coated latex”

iii)               Lines 183-185: a little bit reiterative use of “as the size of”. Please, change it slightly.

iv)               Line 187: I think that “surface roughness” should be omitted.

v)                  Line 234-5: I consider that present tense could be more adequate.

vi)               Line 236. I would recommend “In addition of” instead of “Despite”

b)      Coating process: Which is the concentration of the silver salt solution? “After incubation for 5 days…” Just in the salt solution? Or there was something else?

Why were those times (5 days +1 day with the Eucalyptus solution) chosen?

c)      AFM measurements: The roughness indicated, is the RMS (root mean square) one or the peak to valley? Seeing the figures, I think that it is the second one. In any case, the roughness should be indicated for both samples and which kind of roughness is used should be clearly stated.

d)      Figure 2 (and many more, indeed): the resolution of the figures is not good enough. In fact, I am not able to discern which elements are found. By the way, which is the color legend for the elemental mapping?

e)      Antibacterial activity. A little bit more of discussion (comparing with other articles) would be advisory.

f)       Table 1. First of all, how has it been measured? I presume that using an optical microscope, but that should be indicated. Additionally, the error should be restricted to just one significant figure (and the results should be changed accordingly)

g)      Figure 4. Another example of low resolution, making more difficult to follow the results.

h)      Figure 5. In addition of improving the resolution, the authors should write (a) and (b) in the figure.

i)       Line 293. A reference should be added about that model, if possible

j) SD = Standard deviation, I presume. That should be clearly indicated the first time it appears.

k)  Figure 4 and 7. The labels of the figures are too long and difficult to read.

Author Response

Reviewer 1

In the submitted manuscript, the authors have proposed a green synthesis method of Ag NPs on urinary catheters and have characterized them thoroughly, with special focus on the antimicrobial properties. The work is well suited for the journal and interesting enough for its publication but there are several issues to be addressed before I recommend its publication. In particular, my commentaries are the following ones:

  1. a) There are several grammar errors and/or phrases which meaning is not totally clear. Please, check the grammar carefully. I put here several examples.

Author response

Manuscript was checked.

  1. Line 55-56: “…have been modified by fabricating with AgNPs using different approaches…” I think that the “with” should be omitted.

Author response

Text edited (Line 73).

  1. Line 81: I am not sure of the meaning of “…balloon catheter was used silicone-coated latex”

Author response

Text modified (Line 98-99).

Commercial silicone-coated latex Foley catheters were used in this study.

  • Lines 183-185: a little bit reiterative use of “as the size of”. Please, change it slightly.

Author response

Text edited (Line 206-207).

  1. Line 187: I think that “surface roughness” should be omitted.

Author response

Text edited (Line 206).

  1. Line 234-5: I consider that present tense could be more adequate.

Author response

Text edited (Line 263-265).

Silver ions inhibit activities of several enzymes involved in various important metabolic pathways of bacteria including glycolysis pathway, oxidative stress homeostasis, and the pentose phosphate pathway, resulting in cell death.

  1. Line 236. I would recommend “In addition of” instead of “Despite”

Author response

Text edited (Line 265).

  1. b) Coating process: Which is the concentration of the silver salt solution?

Author response

Information added (Line 105).

“After incubation for 5 days…” Just in the salt solution? Or there was something else?

Author response

Rewritten (Line 104-110).

Foley urinary catheters were cut into small 0.5 cm segments. The segments were rinsed with deionized water and then immersed in a silver salt solution (0.1 g/mL) for 5 days under a dark condition. The samples were washed twice with deionized water and E. camaldulensis extract was added, left at room temperature for further 24 h. To obtain the AgNPs-coated urinary catheters, the samples were removed from the extract, washed with deionized water, and air-dried.

Why were those times (5 days +1 day with the Eucalyptus solution) chosen?

Author response

Rewritten (Line 104-110).

  1. c) AFM measurements: The roughness indicated, is the RMS (root mean square) one or the peak to valley? Seeing the figures, I think that it is the second one. In any case, the roughness should be indicated for both samples and which kind of roughness is used should be clearly stated.

Author response

Information added (Line 184-186).

An uncoated catheter showed a smooth surface with root mean square of 78 nm while AgNPs-coated Foley catheter demonstrated a surface roughness of approximately 199 nm.

  1. d) Figure 2 (and many more, indeed): the resolution of the figures is not good enough. In fact, I am not able to discern which elements are found. By the way, which is the color legend for the elemental mapping?

Author response

Table S1 were added (Supplementary information).

All figures were checked.

  1. e) Antibacterial activity. A little bit more of discussion (comparing with other articles) would be advisory.

Author response

Information added (Line 224-229).

A modified Kirby-Bauer method has been extensively used to screen antibacterial activity of several antimicrobial compounds and antimicrobial-coated medical devices. Silicon catheter coated with Pistacia lentiscus-mediated synthesized AgNPs could inhibit both Gram-positive and Gram-negative bacteria [12]. Similarly, others have demonstrated bio-inspired antimicrobial coating against S. aureus and E. coli [13].

  1. Goda, R.M., El-Baz, A.M., Khalaf, E.M., Alharbi, N.K., Elkhooly, T.A., & Shohayeb, M.M. (2022). Combating bacterial biofilm formation in urinary catheter by green silver nanoparticle. Antibiotics (Basel). 11, 495.
  2. Yassin, M.A., Elkhooly, T.A., Elsherbiny, S.M., Reicha, F.M., Shokeir, A.A. (2019). Facile coating of urinary catheter with bio–inspired antibacterial coating. Heliyon, 5, e02986.

  1. f) Table 1. First of all, how has it been measured? I presume that using an optical microscope, but that should be indicated. Additionally, the error should be restricted to just one significant figure (and the results should be changed accordingly)

Author response

Information added (Line 135-137, 234-235).

  1. g) Figure 4. Another example of low resolution, making more difficult to follow the results.

Author response

Figure 4 edited

  1. h) Figure 5. In addition of improving the resolution, the authors should write (a) and (b) in the figure.

Author response

Figure 5 edited

  1. i) Line 293. A reference should be added about that model, if possible

Author response

Reference added (Line 324).

[10] Zhang, S., Liang, X., Gadd, G. M., & Zhao, Q. (2019). Superhydrophobic coatings for urinary catheters to delay bacterial biofilm formation and catheter-associated urinary tract infection. ACS Applied Bio Materials, 3, 282-291.

  1. j) SD = Standard deviation, I presume. That should be clearly indicated the first time it appears.

Author response

Information added (Line 136-137).

  1. k) Figure 4 and 7. The labels of the figures are too long and difficult to read.

Author response

Figure 4 and 7 edited

Reviewer 2 Report

The paper "Eucalyptus-mediated synthesized silver nanoparticles-coated urinary catheter inhibits microbial migration and biofilm formation" presents a well-designed study on the microbial contamination of a specific medical device class, namely urinary catheters.
Nevertheless, as the authors focused on coating the catheters with green silver nanoparticles, the Introduction section should be extended,  as significant research in green nano-Ag, NPS, and their applications have been made.
In the meantime, the Conclusion section should be more comprehensive and extended based on the good results obtained.
Some of the English expressions should be revised. For instance, in line 234, "Silver ion abolished activities of several..." the usage of abolished might not be appropriate.
Also, please use mL instead of ml as a measuring unit.
For the diagrams in Figure 4 or Figure 7, it is necessary to re-write the legend on OY axes (similar to Figure 8).
The references section is scarce regarding the green Ag NPS.

Author Response

Reviewer 2

The paper "Eucalyptus-mediated synthesized silver nanoparticles-coated urinary catheter inhibits microbial migration and biofilm formation" presents a well-designed study on the microbial contamination of a specific medical device class, namely urinary catheters.

Nevertheless, as the authors focused on coating the catheters with green silver nanoparticles, the Introduction section should be extended, as significant research in green nano-Ag, NPS, and their applications have been made.

Author response

Information added (Line 59-70).

Several approaches for the synthesis of AgNPs have been reported. A synthesis method using plant extracts as reducing and stabilizing agents is increasing interest as it is simple, environmentally friendly, and inexpensive. Our previous studies demonstrated that the green synthesis of AgNPs using leaf extracts from plants in the Myrtaceae family, such as Eucalyptus sp., present strong antimicrobial activity [3–4]. Eucalyptus camaldulensis is one of the most common species because it is widely cultivated in many countries, including Thailand, for paper industries. In addition, E. camaldulensis leaf extract is composed of several phytochemicals such as polyphenols, carboxylic acids, and proteins, that may help for reducing Ag+ to Ag0 [4]. Green synthesized AgNPs have been used as an antimicrobial coating compound for many medical devices including, endotracheal tubes, titanium implants, catheters, surgical sutures, and textiles [5–9].

In the meantime, the Conclusion section should be more comprehensive and extended based on the good results obtained.

Author response

Text edited (Line 368-383).

Eucalyptus-mediated synthesized AgNPs coated Foley urinary catheters provided nano-rough surfaces. The biosynthesized AgNPs were found both on the catheter surfaces and inside the material substrate. AgNPs-coated Foley catheters exhibited broad-spectrum antimicrobial activity against important pathogens causing CAUTIs. The coated catheters demonstrated effective antimicrobial adhesion and antibiofilm formation in the culture medium supplemented with human urine. There was no cytotoxic effect of the coated catheters on HeLa cells. In vitro models mimicking the pathogenesis of CAUTIs revealed that the coated urinary catheters were able to inhibit bacterial migration from the contaminated sites. The promising results demonstrated that AgNPs-coated urinary catheters could be applied to catheterized patients to prevent microbial adhesion, a vital step for developing infections

Some of the English expressions should be revised. For instance, in line 234, "Silver ion abolished activities of several..." the usage of abolished might not be appropriate.

Author response

Text edited (Line 263-265).

Silver ions inhibit activities of several enzymes involved in various important metabolic pathways of bacteria including glycolysis pathway, oxidative stress homeostasis, and the pentose phosphate pathway, resulting in cell death.

Also, please use mL instead of ml as a measuring unit.

Author response

All unit were checked.

For the diagrams in Figure 4 or Figure 7, it is necessary to re-write the legend on OY axes (similar to Figure 8).

Author response

Figure 4 and 7 edited.

The references section is scarce regarding the green AgNPS.

Author response

References added (Line 59-70).

Reviewer 3 Report

The figures should very clear, Also the text of figures 1-6 is not readable., 

Need high-resolution SEM images, color mapping, and EDX to confirm the Ag. 

In Figure 5,6 not denoted as A and B.  And captions should rewrite about figures. 

The figure-7, the caption is misplaced. 

Figure-6a, antibacterials test plate images are very dull, not getting information. need distinct images.  Why coated are shows some bacterial growth zone in every plate. 

Author Response

Response to the Reviews

Reviewer 3

The figures should very clear, Also the text of figures 1-6 is not readable.,

Need high-resolution SEM images, color mapping, and EDX to confirm the Ag.

In Figure 5,6 not denoted as A and B.  And captions should rewrite about figures.

The figure-7, the caption is misplaced.

Author response

All figures edited

Table S1 were added (Supplementary information).

Figure-6a, antibacterials test plate images are very dull, not getting information. need distinct images. Why coated are shows some bacterial growth zone in every plate.

Author response

Figure 6 edited

Figure 6. (A) Catheter bridge model of Proteus mirabilis migrating over 1 cm sections of uncoated and AgNPs-coated urinary catheter after 24, 48, and 72 h. Arrow demonstrated bacterial migration from right to left side. (B) The distance of the bacterial migration values. The values indicate mean ± SD from three independent experiments performed in triplicate, p < 0.05. A representative photograph was from one of three independent examinations.

Round 2

Reviewer 1 Report

The article has improved greatly after the corrections and I recommend its publication. Nevertheless, I have several comments which should be taken into account:

  a)  The authors have not explained me why those periods of time (5 days + 1 day) were chose, which could be commented directly in the response

b b) Line 136-138: “The diameter of the clear zone was measured using a vernier caliper 136 which is precise up to two decimal places. The results were mean ± standard deviation 137(SD) from three independent experiments performed in triplicate.” and table 1

Even if the error of the caliper is in the second decimal, if the standard deviation is in the range of the first decimal, the error is in the first decimal, not in the second (the experimental error is the greater of both). Therefore, that should be corrected (e.g., instead of 10.58 ± 0.14, it should be 10.6 ± 0.1).

 c c) By the way, which is the color legend for the elemental mapping? (supplementary information)

Author Response

Response to the Reviews

Reviewer 1

The article has improved greatly after the corrections and I recommend its publication. Nevertheless, I have several comments which should be taken into account:

  1. a) The authors have not explained me why those periods of time (5 days + 1 day) were chose, which could be commented directly in the response

Author response

Preliminary trials on several conditions and timing to impregnate AgNPs on catheter surfaces were studied and we found that those incubation times provided the best antibacterial activity after screening by zone inhibition assay (data not shown). Therefore, coating catheter in silver solution for 5 days, followed by eucalyptus solution for 1 day was selected for further physical characterization and biological investigation.

  1. b) Line 136-138: “The diameter of the clear zone was measured using a vernier caliper 136 which is precise up to two decimal places. The results were mean ± standard deviation 137(SD) from three independent experiments performed in triplicate.” and table 1

Even if the error of the caliper is in the second decimal, if the standard deviation is in the range of the first decimal, the error is in the first decimal, not in the second (the experimental error is the greater of both). Therefore, that should be corrected (e.g., instead of 10.58 ± 0.14, it should be 10.6 ± 0.1).

Author response

The results were corrected suggested.

  1. c) By the way, which is the color legend for the elemental mapping? (supplementary information)

Author response

Color legend added.

Reviewer 3 Report

1. The figures should very clear, Also the text of figures 1-6 is not readable.

> Still not clear.

2. Need high-resolution SEM images, color mapping, and EDX to confirm the Ag. 

> authors did not provide any clarification about the questions. It is very important to for this manuscript. Authors added new Unclear SEM mapping in Figure 2, but that is not very much important. 

3.  In Figure 5,6 not denoted as A and B.  And captions should rewrite about figures. 

The figure-7, the caption is misplaced. 

> Now figure no changes not mentioned in the letter. 

4. Figure-6a, antibacterials test plate images are very dull, not getting information. need distinct images.  Why coated are shows some bacterial growth zone in every plate. 

> still not clear using same. 

Author Response

Reviewer 3

  1. The figures should very clear, Also the text of figures 1-6 is not readable.

> Still not clear.

Author response

All figures were carefully attended and replaced with high-resolution figures.

  1. Need high-resolution SEM images, color mapping, and EDX to confirm the Ag.

> authors did not provide any clarification about the questions. It is very important to for this manuscript. Authors added new Unclear SEM mapping in Figure 2, but that is not very much important.

Author response

SEM images, color mapping, and EDX to confirm the Ag was added as a figure S1 (supplementary information).

  1. In Figure 5,6 not denoted as A and B. And captions should rewrite about figures.

The figure-7, the caption is misplaced.

> Now figure no changes not mentioned in the letter.

Author response

Figures and figure captions were edited.

  1. Figure-6a, antibacterials test plate images are very dull, not getting information. need distinct images. Why coated are shows some bacterial growth zone in every plate.

> still not clear using same.

Author response

The figure 6 is not zone of inhibition. The experiment was performed to investigate whether the coated catheter can restrict bacterial migration through the catheter lumen. Bacterial inoculum was dropped on one side of the catheter piece. In this side, the bacteria had to grow as a control. However, the coated catheter could stop the migration of the tested bacteria as there was no bacteria on another side of the catheter.
